# Molecular investigation of *TSHR* gene in Bangladeshi congenital hypothyroid patients

**Mst. Noorjahan Begum[1,2,3], Rumana Mahtarin[1,4], Md. Tarikul Islam [1], Sinthyia Ahmed[5], Tasnia Kawsar Konika[6], Kaiissar Mannoor[1], Sharif Akhteruzzaman[2], Firdausi Qadri [1,7]***

**1** Institute for Developing Science and Health Initiatives (ideSHi), ECB Chattar, Mirpur, Dhaka, Bangladesh, **2** Department of Genetic Engineering & Biotechnology, University of Dhaka, Dhaka, Bangladesh, **3** Virology Laboratory, Infectious Diseases Division, International Centre for Diarrhoeal Disease Research, Bangladesh, Mohakhali, Dhaka, Bangladesh, **4** Department of Biochemistry and Molecular Biology, Shahjalal University of Science and Technology, Sylhet, Bangladesh, **5** Division of Computer Aided Drug Design, The Red-Green Research Centre, BICCB, Tejgaon, Dhaka, Bangladesh, **6** Nuclear Medicine and Allied Sciences, Bangabandhu Sheikh Mujib Medical University (BSMMU), Shahbag, Dhaka, Bangladesh, **7** Mucosal Immunology and Vaccinology, Infectious Diseases Division, International Centre for Diarrhoeal Disease Research, Bangladesh, Mohakhali, Dhaka, Bangladesh

* fqadri333@gmail.com

**Data Availability Statement:** Relevant data of the study findings have been shared in figshare. Link: https://figshare.com/s/b10e48d2a8e110cca57e.

## Abstract

The disorder of thyroid gland development or thyroid dysgenesis accounts for 80–85% of congenital hypothyroidism (CH) cases. Mutations in the *TSHR* gene are mostly associated with thyroid dysgenesis, and prevent or disrupt normal development of the gland. There is limited data available on the genetic spectrum of congenital hypothyroid children in Bangladesh. Thus, an understanding of the molecular aetiology of thyroid dysgenesis is a prerequisite. The aim of the study was to investigate the effect of mutations in the *TSHR* gene on the small molecule thyrogenic drug-binding site of the protein. We identified two nonsynonymous mutations (p.Ser508Leu, p.Glu727Asp) in the exon 10 of the TSHR gene in 21 patients with dysgenesis by sequencing-based analysis. Later, the $TSHR_{368-764}$ protein was modeled by the I-TASSER server for wild-type and mutant structures. The model proteins were targeted by thyrogenic drugs, MS437 and MS438 to perceive the effect of mutations. The damaging effect in drug-protein complexes of mutants was explored by molecular docking and molecular dynamics simulations. The binding affinity of wild-type protein was much higher than the mutant cases for both of the drug ligands (MS437 and MS438). Molecular dynamics simulates the dynamic behavior of wild-type and mutant complexes. MS437-$TSHR_{368-764}$MT2 and MS438-$TSHR_{368-764}$MT1 showed stable conformations in biological environments. Finally, Principle Component Analysis revealed structural and energy profile discrepancies. $TSHR_{368-764}$MT1 exhibited much more variations than $TSHR_{368-764}$WT and $TSHR_{368-764}$MT2, emphasizing a more damaging pattern in $TSHR_{368-764}$MT1. This genetic study might be helpful to explore the mutational impact on drug binding sites of TSHR protein which is important for future drug design and selection for the treatment of congenital hypothyroid children with dysgenesis.

**Funding:** This study was partially funded by a grant from the University of Dhaka received from the University Grants Commission (CP-4029) for Higher Education Quality Enhancement Project. The first author was a Ph.D. student (from 2016 to 2019) under a grant from the University of Dhaka. The Ph.D. program was finished in December 2019. The funders had no role in study design, data collection and analysis, decision to publish, or preparation of the manuscript.

**Competing interests:** The authors have declared that no competing interests exist.

## 1. Introduction

Congenital hypothyroidism is associated with various factors including genetics. Genetic causes account for about 15 to 20 percent of congenital hypothyroidism (CH). Although CH is a genetically heterogeneous disorder, the candidate genes divide the disorder into two main groups namely thyroid dysgenesis and thyroid dyshormonogenesis. Different studies including online databases such as Genetics Home Reference and Online Mendelian Inheritance in Men (OMIM) suggested that about 10–20 percent of total cases with CH were associated with thyroid dyshormonogenesis that would result from mutations in one of several genes involved in the biosynthesis of thyroid hormones [1]. The above-mentioned databases also described that about 80–85% of CH cases are associated with disorders of thyroid gland development (Dysgenesis) which is categorized as ectopic (located in a distant region, 40%), agenesis (absent of thyroid gland, 40%), and the other cases are accompanying with hypoplasia (small size). Although the actual cause of thyroid dysgenesis is still under investigation, some studies have suggested that 4 major genes that play roles in the proper growth and development of the thyroid gland, such as *TSHR* (Thyroid 3 stimulating hormone receptor) and three transcription factors- *TTF-1*, *TTF-2*, and *PAX8* (paired box-8, transcription factor) [2]. Mutations in these genes prevent or disrupt the normal development of the gland. The *TSHR* gene is predominantly related to thyroid dysgenesis, as most of the mutations occurs in the gene in CH patients [2]. TSHR is a G protein-coupled transmembrane receptor which is present on the surface of thyroid follicular cells. TSH, secreted by the anterior pituitary, mediates its effect through TSHR which is crucial for thyroid gland development and function. The *TSHR* gene is located on chromosome 14q31 and contains 11 exons code for a receptor protein of 764 amino acid residues [3, 4]. TSHR has high affinity binding sites for TSH. Mutations in the *TSHR* gene result in mutant TSHR protein which lacks its binding affinity to TSH or loses its ability to activate adenylate cyclase. Thus, mutant TSHR protein disrupts thyroid gland development and proper functioning. *TSHR* mutation may also be present in a normally placed thyroid gland. *TSHR* gene mutation is reported to be inherited as an autosomal recessive manner and exon 10 is known to carry the majority of the mutations [5]. In a high-throughput screening system, two small molecule agonists (MS437 and MS438) exhibited pharmacotherapeutic potential with the highest potency (EC50 of $13 \times 10^{-8}$ M, and EC50 of $5.3 \times 10^{-8}$ M respectively) [6].

Very limited data are available on genetic study of Bangladeshi hypothyroid patients. Therefore, the present study tried to explore the effect of two non-synonymous mutations in the 3D structure of $TSHR_{368-764}$ targeted by thyrogenic drugs, MS437 and MS438 which will help to update any future treatment strategy including suitable drug design for Congenital Hypothyroid children.

## 2. Methods and materials

### 2.1. Study design, clinical settings, and ethical clearance

The study was designed and carried out on 21 confirmed cases of Congenital Hypothyroid children with dysgenesis who were kept under treatment of Levothyroxine (LT4) drug in the Department of Endocrinology and National Institute of Nuclear Medicine and Allied Sciences (NINMAS) of Bangabandhu Shaikh Mujib Medical University (BSMMU). Ethical permission was obtained from the Ethical Review Committee of University of Dhaka (CP-4029) and the study was collaborated with NINMAS and Dept. of Endocrinology, BSMMU for specimen collection. Prior to enrollment of study participants, a written informed consent along with the clinical information was collected from the parent(s) or legal guardian(s) of each patient.

## 2.2. Collection and processing of blood specimens

Blood Specimens were collected from the participants to conduct the molecular, biochemical and metabolic profiling tests. A total of 3ml blood was collected from each participant. All the samples were transported to the laboratory immediately. After the genomic DNA isolation, EDTA containing blood was stored at -70˚C freezer.

## 2.3. Molecular analysis of *TSHR* gene

Now-a-days, gene-based study plays the key role to explore the actual cause of a particular disease. The present study was designed to perform the molecular analysis in various steps.

**2.3.1. Genomic DNA isolation to perform PCR.** Genomic DNA was isolated from the EDTA blood by using Qiagen DNAeasy mini kit according to the manufacturer's instruction. 500 µl of FG1 buffer was taken in a 1.5 ml microcentrifuge tube. 200 µl of whole blood was added to the FG1 buffer and mixed by inverting the tube 5–10 times. The mixture was then centrifuged at 10,000×g for 5 minutes in fixed angle rotor. The supernatant was carefully removed so that the pellet remained in the tube. 1µl of QIAGEN protease was added to 100 µl of FG2 buffer and mixed by vortex in a fresh Eppendorf tube. Then 100 µL of FG2/QIAGEN protease was added to the pellet and vortexed immediately until the pellet was completely dissolved and the color was changed into olive green so that all the protein components were degraded. The mixture was then incubated in a water bath or heat block at 65˚C for 5 minutes. After incubation, 100 µl of isopropanol (100%) was added and mixed by inversion until DNA was precipitated as visible threads. The tube was centrifuged for 5 minutes at 10000×g. The supernatant was discarded and the pellet was dried by keeping the tube inverted state on a clean tissue paper for one minute. 100 µl of 70% ethanol was added and vortexed for 5 seconds. The tube was centrifuged for 5 minutes at 10000×g. The supernatant was carefully aspirated using a micropipette and keeping the micro-centrifuge tube in the inverted state on the tissue paper to allow the pellet to air dry for at least 5 minutes. Over-drying was avoided as the process can make it difficult to dissolve the DNA. Depending on the pellet size, 25–50 µl of nuclease free water was added and the tube was vortexed for 5 seconds and the mixture was incubated at 65˚C for one hour in water bath for dissolving DNA or left overnight at room temperature. Finally, the concentration and the purity of the DNA was measured using a Nano drop machine and adjusted the concentration for PCR.

**2.3.2. Polymerase Chain Reaction (PCR) amplification of *TSHR* gene.** The isolated DNA was then amplified by PCR using *TSHR* gene-specific primers. At first, we performed PCR using primers set that could flank the sequence between exon 1 to exon 10, since global data showed that most of the common mutations in the *TSHR* gene of the patients with Congenital Hypothyroidism were confined in this region. Next, we conducted PCR for other regions of the *TSHR* gene. The primer sequences are listed in the Table 1 as follows. To amplify the desired target sequence of *TSHR* gene, PCR amplification was conducted on a thermal cycler (Bio-Rad, USA). The final reaction volume was 10 µl for each of the reactions which contained 1 µL 10X PCR buffer, 0.3 µL 25mM $MgCl_2$, 2 µL 5X Q-solution, 1.6 µL 2.5 mM dNTPs mixture, 0.2 µL 10mM Forward primer and 0.2 µL Reverse primer, 0.05 µL Taq DNA Polymerase, 50 ng of genomic DNA and total reaction volume was made up to 10µL by addition of nuclease free water. The thermal cycling condition included (a) initial denaturation at 95˚C for 5 minutes, (b) cyclic denaturation at 95˚C for 40 seconds and annealing at 58˚C for 35 seconds and extension at 72˚C for 40 seconds; and (c) final extension at 72˚C for 5 minutes for 35 cycles.

**2.3.3. Sanger Sequencing of PCR products.** Before sequencing, the PCR products were purified using a Qiagen PCR purification kit (Qiagen) following manufacturer's instruction.

**Table 1. List of primers for PCR amplification and Sanger sequencing of *TSHR* gene.**

| Primer name | Primer sequence | Tm | Product size |
| --- | --- | --- | --- |
| TSHR_Ex1F | GGCATCTAAACTAGGCTTTGGAG | 62.6°C | 646 bp |
| TSHR_Ex1R | CTTCGGGCTGTTATTGAGCTGC | 65.2°C | |
| TSHR_Ex2F | AGTGTGATGCGAGGCAAGAC | 64.3°C | 645 bp |
| TSHR_Ex2R | CAGCTAAGGTTTTGCCATATCCC | 63.1°C | |
| TSHR_Ex3F | GTGGAACATTCCACAGGGTGAC | 64.7°C | 622 bp |
| TSHR_Ex3R | CTTCCAACCATGGAATTGAGGTG | 63.3°C | |
| TSHR_Ex4F | AAAGTGGACAGAAACCAAGCC | 62.9°C | 470 bp |
| TSHR_Ex4R | GATCATTTCACCCGATACCTTGC | 63°C | |
| TSHR_Ex5F | CCGAGCAGATGTATTGACACCAG | 64.3°C | 577 bp |
| TSHR_Ex5R | TACCCAAGTCTCTCTTGAGCC | 62.7°C | |
| TSHR_Ex6F | TCCAGGTGCATGTCATCTAGG | 63.1°C | 556 bp |
| TSHR_Ex6R | GGTTGCATGGTCTGTAATGCC | 63.4°C | |
| TSHR_Ex7F | AGAGACTGCAGCTGCTCCTCC | 67.2°C | 646 bp |
| TSHR_Ex7R | AGCTTTGGAACTTACCATTGGAG | 62.7°C | |
| TSHR_Ex8F | GAATGTTTTAAGTGCTCAAGCCAG | 62.4°C | 834 bp |
| TSHR_Ex8R | GGCAATGATACAGAGGCTTCAGG | 65.2°C | |
| TSHR_Ex9F | AGCATTTGTACTACTGGATACTGG | 61.8°C | 533 bp |
| TSHR_Ex9R | CTTCCAATTTCCTCTCCACCTG | 62.3°C | |
| TSHR_Ex10F1 | AGGAATGATGTCACAGAAACAGGC | 64.7°C | 1151 bp |
| TSHR_Ex10R1 | GTGATGGCATACCAGCGCTCCAG | 68.4°C | |
| TSHR_Ex10F2 | CGCTTTCTCATGTGCAACCTGGC | 67.7°C | 1030 bp |
| TSHR_Ex10R2 | GGGTGTCATGGGATTGGAATGC | 65.2°C | |

The cycle sequencing PCR was then performed by BigDye Chain Terminator version 3.1 Cycle Sequencing Kit (Applied Biosystems, USA) applying manufacturer's instructions. The thermal cycling profile comprised (i) initial denaturation at 94°C for 1 minute, (ii) 25 cycles of denaturation at 94°C for 10 seconds, annealing at 58°C for 5 seconds and extension at 60°C for 4 minutes, and (iii) final extension at 60°C for 10 minutes. After cycle sequencing PCR, the products were purified using BigDye XTerminator® Purification Kit (Applied Biosystems). Then, sequencing of the purified cycle sequencing products was executed on the ABI PRISM 310 automated sequencer (Applied Biosystems, USA) [7].

**2.3.4. Sequencing data analysis.** The Sequencing data were obtained from ABI PRISM 310 data collection software version 3.1.0. FASTA format of sequencing data were utilized to identify mutations in the *TSHR* gene by alignment with the reference sequence (Accession number; NG_009206.1 retrieved from the NCBI database) through the basic local alignment search tool (BLAST). The nucleotides sequence was converted into corresponding amino acids by ExPASy translate tool [7].

## 2.4. Prediction of 3D structure of TSHR protein and ligand selection

After performing the Sanger Sequencing, we detected two mutations, one was in the transmembrane (TM)-domain and another was in cytoplasmic (CT)-domain of TSHR protein. TSHR protein is composed of a total of 764 amino acids where the TM-domain and CT-domain belong to 368 to 764 amino acids of the full length TSHR protein. I-TASSER server was used to predict the 3D structures of wildtype and mutant TSHR protein (TM and CT domains) due to lack of the full-length experimental structure. I-TASSER provided the five models for $TSHR_{368-764}$ based on the detected template rhodopsin x-ray crystal structure

(PDB:1F88) [6] by LOMETS (local meta-threading-server) from the PDB library [8]. Target-template alignment was also provided for each model structure. On the basis of Confidence score (C-score), template modelling (TM) score, root mean square deviation (RMSD) score, and target-template aligned structures, we obtained best model for each structure. The I-TASSER predicted structure was compared with AlphaFold predicted structure by TM-align algorithm which detects the best structural alignment (https://zhanggroup.org/TM-align/) [9]. Two promising small molecule ligands (MS437 and MS 438) were selected which act as agonist to TSHR protein. Since among the small molecules MS437 interacts with threonine 501 (T501), and MS438 interacts with residues serine 505 (S505) and glutamic acid 506 (E506) bind to the intrahelical region of TM3 of TSHR protein [6], we targeted the region to see the effect of mutations on that particular site.

## 2.5 Molecular docking, protein-ligand interactions, and molecular dynamics (MD) simulation

Finally, the molecular docking was performed for I-TASSER predicted and AlphaFold predicted wild-type and mutant proteins using PyRx software [10, 13, 14]. Grid box center was x = 72.5922; y = 72.4245; z = 72.6927 and Grid box size was 25 in every axis during docking encompassing the active site residues Thr134 (501) for MS437, and Ser138 (505) and Glu139 (506) for MS438 in the intrahelical region of TM3 of TSHR protein. The binding affinity for both I-TASSER predicted and AlphaFold predicted structures was analyzed using PyRx software and PRODIGY [11, 12]. Non-covalent interactions were also observed both for of MS437 and MS438 molecules using BIOVIA Discovery Studio version 4.5.

The MD simulation was implemented through YASARA suits [13] employing AMBER14 force field [14] for calculations. The membrane was built during simulation. YASARA scanned the plausible transmembrane region comprising hydrophobic residues among the secondary structure elements of proteins. The protein was projected to the membrane, YASARA presented the recommended membrane insertion with required size (69.2 Å × 7.3 Å) containing phosphatidyl-ethanolamine, -choline, and -serine lipid constituents. The whole simulation system was equilibrated for 250 ps. During the equilibration phase, membrane was artificially stabilized. The entire environment was equilibrated at 310K temperature with 0.9% NaCl and water solvent. The temperature was controlled by Berendsen thermostat process during simulation. The particle Mesh Ewald algorithm maintained long-range electrostatic interactions. The periodic boundary condition was applied for the whole simulation. The time step was 1.25 fs during 50 ns MD simulation. The snapshots were collected at every 100 ps [12]. Diverse data containing root mean square deviation (RMSD), root mean square fluctuation (RMSF), total number of hydrogen bonds, radius of gyration, solvent-accessible surface area (SASA), and molecular surface area (MolSA) were retrieved from MD simulations, following our earlier MD data analysis [13, 14].

## 2.6. Principal Component Analysis (PCA)

MD simulation data were applied to explore the structural and energy variabilities via principal component analysis (PCA) among TSHR-small molecule ligand (MS437 and MS438) complexes. The different multivariate energy factors were employed to explore the existing variability in the MD trajectory applying the low-dimensional space [13]. The variables from MD data were bond distances, bond angles, dihedral angles, planarity, van der Waals energies, and electrostatic energies considered for the structural and energy factors [14]. The data pre-processing were implemented by centering and scaling. In the analysis, final 45 ns MD trajectories were applied for exploration of the variations. The PCA model is reproduced by the following

equation:

$$X = T_k P_k^T + E$$

Where, multivariate factors are presented into the resultant of two new matrices by $X$ matrix i.e. $T_k$ and $P_k$; $T_k$ matrix of scores correlates the samples; $P_k$ matrix of loadings associates the variables, $k$ is the number of factors available in the model and $E$ demonstrates the matrix of residuals. The trajectories were analysed through R, RStudio and essential codes. The R package ggplot2 was utilized for PCA plots generation.

## 3. Result

### 3.1. Investigation of mutation in *TSHR* gene

All the 21 patients with dysgenesis had mutation in exon 10 among a total of 11 exons in *TSHR* gene. The mutations we found namely, c.1523C>T (p.Ser508Leu) and c.2181G>C (p.Glu727Asp). Among the 21 patients, only one patient had mutation c.1523C>T (p.Ser508Leu) and 20 patients had the other variant c.2181G>C (p.Glu727Asp). The variants were then analyzed by bioinformatics tools to explore the pathogenic effect. Firstly, the mutations were tested by Polyphen 2, Mutation taster, and PROVEAN bioinformatics tools to see whether they possessed damaging effect or not. We found that the mutation c.1523C>T probably had damaging effect and c.2181G>C variant showed benign effect. Fig 1 represents a chromatogram showing the mutation (c.1523C>T) for the specific participant and Table 2 shows the mutations found in *TSHR* gene.

### 3.2. Effect of mutations on predicted 3D structure of TSHR protein

The I-TASSER predicted best structures were designated as wild-type TSHR$_{368-764}$WT, p. Ser508Leu variant as TSHR$_{368-764}$ MT1 and p.Glu727Asp variant as TSHR$_{368-764}$ MT2. Table 3

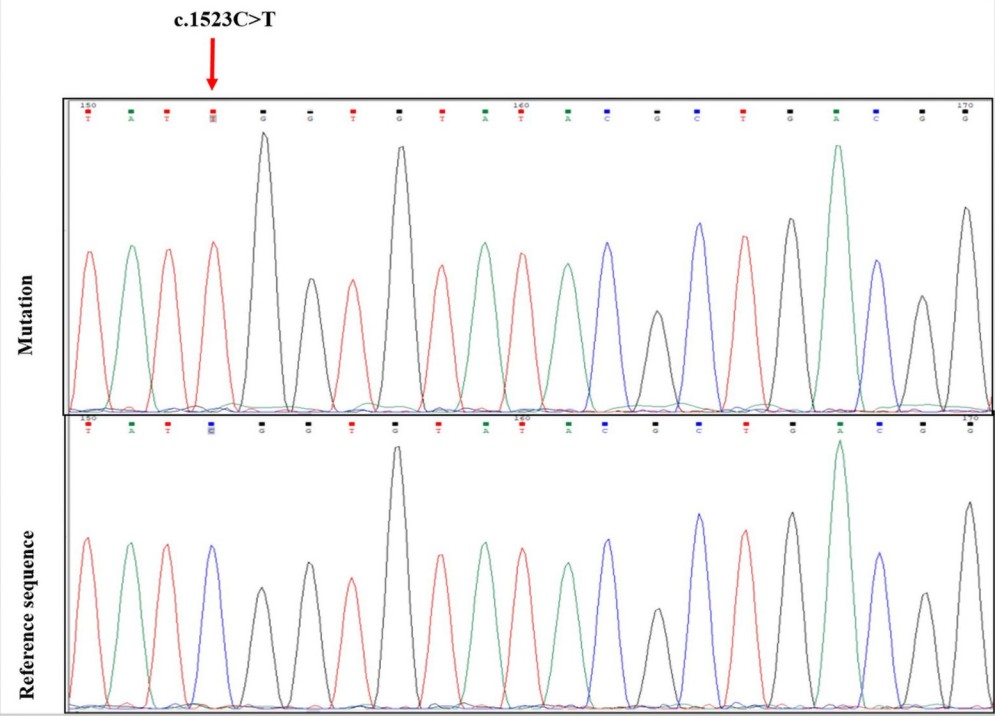

**Fig 1. A representative chromatogram showing a mutation in exon 10 of *TSHR* gene.**

**Table 2. Mutations detected in the *TSHR* gene of hypothyroid patients and analysis of the effect of mutations using different bioinformatics tools.**

| Nucleotide position | Amino acid position | Polyphen 2 | Mutation Taster | PROVEAN |
|---|---|---|---|---|
| **c.1523C>T** | p.Ser508Leu | Probably Damaging | Disease causing | Deleterious |
| **c.2181G>C** | p.Glu727Asp | Benign | Benign | Neutral |

shows the C-score (Confidence score) for wild-type, MT1 and MT2 was -0.71, -0.61 and -0.63, respectively. The TM score (Template Modelling score) and RMSD were 0.62±0.14, 8.4±4.5Å for TSHR$_{368-764}$ WT; 0.64±0.13 and 8.2±4.5Å for TSHR$_{368-764}$ MT1; 0.63±0.13 and 8.2±4.5Å for TSHR$_{368-764}$ MT2. C-score indicated confidence score to assess the global accuracy of predicted models which is calculated based on the significance of threading template alignments and the convergence parameters of the structure assembly simulations. C-score of higher value [−5,2] suggests a model with a high confidence [8]. The global quality of the protein model prediction had been assessed by the TM-score. The TM-score represents the similarity between two protein structures and the accuracy of structure modeling. The TM-score (TM-score > 0.5) of the predicted proteins indicated structures were in correct global topology [15]. From the analysis of TM-score, the target and template (rhodopsin x-ray crystal structure, PDB:1F88) alignments for TSHR$_{368-764}$ WT, TSHR$_{368-764}$ MT1, and TSHR$_{368-764}$ MT2 were 62%, 64%, and 63% respectively. The target-template superimposed structures were displayed in Fig 2A–2C.

Later, I-TASSER predicted structure was compared with AlphaFold predicted structure based on analysis of their TM score (0.68) by TM-align algorithm. TM score (0.68) indicated AlphaFold and I-TASSER predicted structures were in same fold with correct global topologies (TM score>0.5). AlphaFold and I-TASSER both provide highly accurate structures [8, 16, 17] while I-TASSER (https://zhanggroup.org/I-TASSER/) got recognition as the No 1 server for protein structure prediction in community-wide CASP experiments. Hence, I-TASSER predicted models were utilized for further analysis. Moreover, the full length TSHR protein predicted by AlphaFold was shown (Fig 2D) and the structural alignment of TSHR$_{368-764}$ for AlphaFold, and I-TASSER was shown in superimposed pose (Fig 2E). Fig 3 depicts the structures of the TSHR protein and the small molecules MS437 and MS438.

## 3.3. Molecular docking and protein-ligand interactions of MS437 and MS438 with TSHR proteins (wild-type and mutant)

The structures of the small molecules MS437 and MS438 were optimized. After molecular docking best docking poses for the protein-ligand complexes were selected evaluating their binding affinity and interaction. The binding affinities for the small molecules were obtained

**Table 3. Summary of the corresponding model numbers, C–score, TM–score and the RMSD–score of the predicted 3D structures of TSHR$_{368-764}$ WT, TSHR$_{368-764}$ MT1, and TSHR$_{368-764}$ MT2.**

| Features | TSHR$_{368-764}$ WT | TSHR$_{368-764}$ MT1 | TSHR$_{368-764}$ MT2 |
|---|---|---|---|
| Model no. | 01 | 01 | 01 |
| C–score | -0.71 | -0.61 | -0.63 |
| TM–score | 0.62±0.14 | 0.64±0.13 | 0.63±0.13 |
| RMSD (Å) | 8.4±4.5Å | 8.2±4.5Å | 8.2±4.5Å |

C–score = Confidence score range: [−5,2]; TM–score = Template Modelling score, TM–score < 0.17 indicates random similarity and TM–score > 0.5 indicates correct similarity; RMSD = Root Mean Square Deviation.
WT = Wild-type, MT1 = Mutant 1 (p.Ser508Leu) and MT2 = Mutant 2 (p.Glu727Asp).

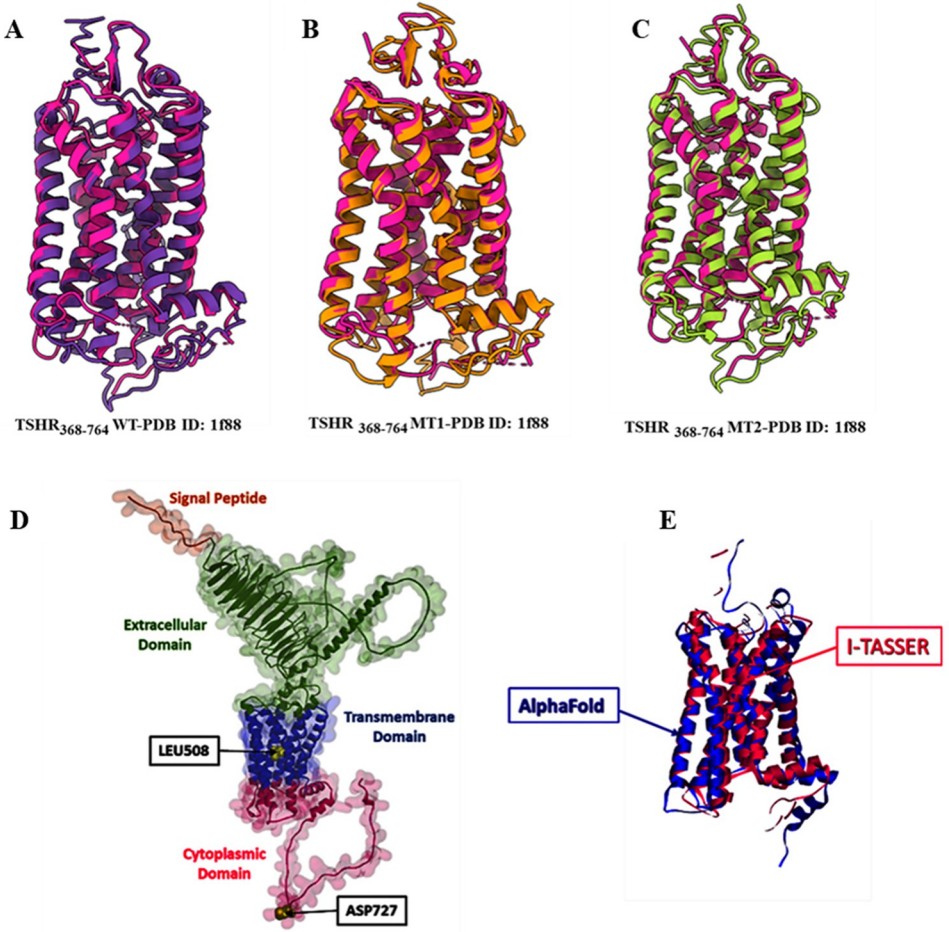

**Fig 2. Comparison of TSHR protein model structures.** Superimposed snapshots of TSHR protein, (A) TSHR$_{368-764}$ WT (purple), (B) TSHR$_{368-764}$ MT1 (orange), and (C) TSHR$_{368-764}$ MT2 (lemon) with PDB:1F88 (Pink). (D) Full length TSHR protein predicted by AlphaFold. Mutations are presented in yellow color. (E) Superimposed snapshot of TM-domain and cytoplasmic domain of TSHR protein predicted by AlphaFold (blue) and I-TASSER (deep Pink).

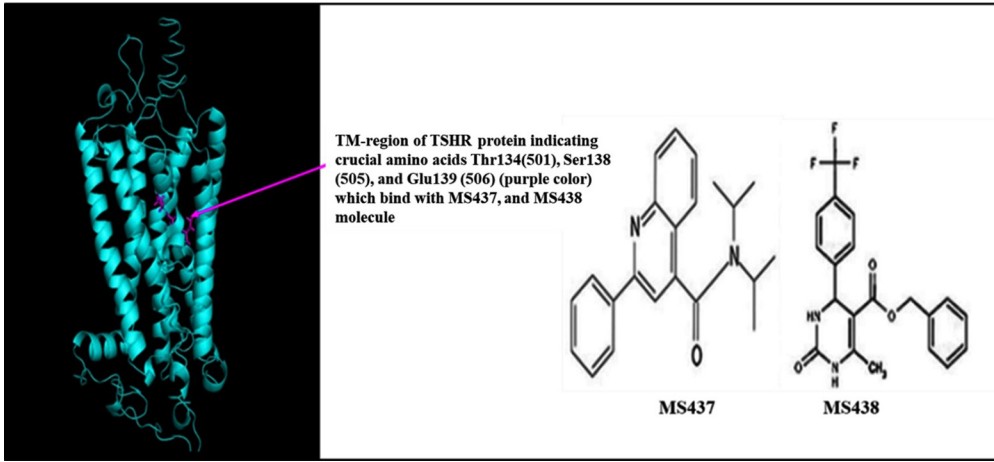

**Fig 3. The structure of TSHR protein (368–764) showing crucial amino acids responsible for binding with MS437 and MS438 molecules.**

**Table 4. Binding Affinity (kcal/mol) of MS437 with TSHR$_{368-764}$ WT, TSHR$_{368-764}$ MT1 and TSHR$_{368-764}$ MT2 proteins, and AlphaFold predicted full-length TSHR proteins after flexible docking.**

| Protein type | PyRx software | PRODIGY |
|---|---|---|
| TSHR$_{368-764}$ WT | -6 | -5.45 |
| TSHR$_{368-764}$ MT1 | -4.8 | -5.28 |
| TSHR$_{368-764}$ MT2 | -5.7 | -5.27 |
| AlphaFold_TSHR-WT | -8.6 | -5.46 |
| AlphaFold_TSHR-MT1 | -8.6 | -5.45 |
| AlphaFold_TSHR-MT2 | -8.4 | -5.41 |

WT = Wild-type; MT1 = Mutant 1 (p.Ser508Leu) and MT2 = Mutant 2 (p.Glu727Asp).

from both PyRx software and PRODIGY. The Table 4 showed that the binding affinities of the wild -type TSHR protein (-6 kcal/mol, -5.45 kcal/mol for TSHR$_{368-764}$WT) were higher compared to the mutant cases (-4.8 kcal/mol, -5.28 kcal/mol for TSHR$_{368-764}$ MT1 and -5.7 kcal/mol, -5.27 kcal/mol for TSHR$_{368-764}$ MT2) in both PyRx software and PRODIGY. Also, the binding affinities for the small molecules were obtained from both PyRx software and PROD-IGY for AlphaFold predicted structures and the values were included in Table 4. Total non-covalent interactions were 11, 19 and 12 for wild-type, MT1 and MT2, respectively (Table 5). MS437 bound to Threonine 501 and MS438 bound to Serine 505 and Glutamic acid 506 of transmembrane helix3 (TMH3) in full length TSHR protein [6] with corresponding amino acid position Thr134, Ser138 and Glu139, respectively in TM-region (Fig 3). We tried to investigate whether these crucial amino acids could interact with small molecule thyrogenic drugs. We found that in case of MS437 none of these three amino acids could interact with both wild-type and mutant cases. On the other hand, MS438 interacted with all the crucial amino acids including Thr134, Ser138 and Glu139 for wild-type case and for the mutant cases (TSHR$_{368-764}$ MT1 and TSHR$_{368-764}$ MT2), it could interact only with Ser138. The binding affinities were -7.1 kcal/mol; -5.59 kcal/mol, -5.4 kcal/mol; -5.77 kcal/mol, and -2.6 kcal/mol; -5.50 kcal/mol for TSHR$_{368-764}$WT, TSHR$_{368-764}$ MT1, and TSHR$_{368-764}$ MT2, in both PyRx software and PRODIGY respectively. In Table 6 the binding affinity of wild-type protein was much higher for MS438 than mutants. Also, the binding affinities for the small molecules were obtained from both PyRx software and PRODIGY for AlphaFold predicted structures and the value were included in Table 6. All the non-covalent interactions (Table 7) were depicted in Figs 4 and 5 for MS437 and MS438 respectively.

**Table 5. Non-covalent interactions of MS437 with TSHR$_{368-764}$ WT, TSHR$_{368-764}$ MT1, and TSHR$_{368-764}$ MT2 proteins after flexible docking.**

| Protein type | Hydrogen bond | Hydrophobic bond | Acceptor | Donor | Total interactions |
|---|---|---|---|---|---|
| TSHR$_{368-764}$ WT | SER274 (641) | VAL219 (586), ALA277 (644), VAL135 (502), MET270 (637), ILE273 (460), PRO204 (571), VAL297 (664) | _ | _ | 11 |
| TSHR$_{368-764}$ MT1 | - | VAL135 (502), LEU222 (589), VAL219 (586), MET270 (637), ILE273 (640), VAL219 (586), ALA277 (644), LEU203 (570), LYS293 (660), VAL297 (664), LYS293 (660), VAL297 (664), PHE218 (585), TYR300 (667) | _ | _ | 19 |
| TSHR$_{368-764}$ MT2 | - | VAL261 (628), LEU310 (677), ILE313 (680), ALA306 (673), LEU302 (669), CYS305 (672), ALA306 (673), PHE309 (676) | _ | _ | 12 |

The amino acid residues and their positions are designated as the three letter abbreviations and the corresponding number; in case of TSHR$_{368-764}$ the amino acid outside the first bracket indicates the position in predicted structure and the amino acid residues within the first bracket indicates the real position in full structure, TSHR$_{1-764}$ protein; WT = Wild-type; MT1 = Mutant 1 (p.Ser508Leu) and MT2 = Mutant 2 (p.Glu727Asp).

**Table 6. Binding Affinity (kcal/mol) of MS438 with TSHR$_{368-764}$ WT, TSHR$_{368-764}$ MT1 and TSHR$_{368-764}$ MT2 proteins, and AlphaFold predicted full-length TSHR proteins after flexible docking.**

| Protein type | PyRx software | PRODIGY |
|---|---|---|
| TSHR$_{368-764}$ WT | -7.1 | -5.59 |
| TSHR$_{368-764}$ MT1 | -5.4 | -5.77 |
| TSHR$_{368-764}$ MT2 | -2.6 | -5.50 |
| AlphaFold_TSHR-WT | -10.5 | -5.55 |
| AlphaFold_TSHR-MT1 | -10.6 | -5.44 |
| AlphaFold_TSHR-MT2 | -9.6 | -5.57 |

WT = Wild-type; MT1 = Mutant 1 (p.Ser508Leu) and MT2 = Mutant 2 (p.Glu727Asp).

## 3.4. Molecular dynamics simulation

MD simulation was performed for each complex of TSHR$_{368-764}$WT, TSHR$_{368-764}$ MT1, and TSHR$_{368-764}$MT2 with two designated drugs (MS437 and MS438) for 50 ns time range.

In case of MS437 (Fig 6), the RMSDs for TSHR$_{368-764}$MT2 (0.973–4.965 Å) displayed less fluctuations for α-carbon atoms than TSHR$_{368-764}$WT (0.906–5.91 Å), and TSHR$_{368-764}$ MT1 (0.939–5.504 Å) in Fig 6A. Thus, suggesting that, comparatively MS437-TSHR$_{368-764}$ MT2 was stable in physiological conditions, while more fluctuations were visible in TSHR$_{368-764}$WT at 43.8 ns (RMSD 5.91 Å) and TSHR$_{368-764}$ MT1 till 26.5 ns (RMSD ∼5.2 Å). The Rg manifested quite similar pattern in TSHR$_{368-764}$ MT1 and TSHR$_{368-764}$ MT2. However, TSHR$_{368-764}$WT exhibited more fluctuations initially up to 5.5 ns and later from 46 ns. The average value remained ∼25.40 Å for the three protein complexes. However, low compactness in ligand-mutant complexes was observed during simulation (Fig 6B). In case of SASA, more deviations were found in TSHR$_{368-764}$WT (18587.716–24360.945 Å$^2$) compared to TSHR$_{368-764}$MT1 (18106.599–22314.102 Å$^2$) and TSHR$_{368-764}$ MT2 (19148.648–22851.2 Å$^2$) complexes. However, TSHR$_{368-764}$ MT1 and TSHR$_{368-764}$ MT2 manifested some deviations at 7–19 ns, and 24–43 ns during simulation. Overall, mutant structures TSHR$_{368-764}$ MT1 and TSHR$_{368-764}$ MT2 were more stable as MS437 bound complexes than TSHR$_{368-764}$WT (Fig 6C).

For MolSA in Fig 6D, TSHR$_{368-764}$WT showed (21208.043–25437.671 Å$^2$) much fluctuations in the whole run than TSHR$_{368-764}$MT1 (21135.321–24580.781 Å$^2$) and TSHR$_{368-764}$

**Table 7. Non-covalent interactions of MS438 with TSHR$_{368-764}$ WT, TSHR$_{368-764}$ MT1 and TSHR$_{368-764}$ MT2 proteins after flexible docking.**

| Protein type | Hydrogen bond | Electrostatic bond | Hydrophobic bond | Acceptor | Donor | Total interactions |
|---|---|---|---|---|---|---|
| TSHR$_{368-764}$ WT | ASN223 (590), VAL135 (502), SER138 (505), GLU139 (506), ASN223 (590), | _ | THR134 (501), VAL142 (509), PRO204 (571), TYR215 (582), MET270 (637), VAL297 (664) | ASN223 (590) (H-acceptor) | _ | 16 |
| TSHR$_{368-764}$ MT1 | SER138 (505), ILE273 (640), VAL135 (502), SER274 (641), CYS202 (569), | LYS293 (660) | LEU100 (467), LEU203 (570), PRO204 (571), TYR215 (582), PHE218 (585), ILE273 (640), ALA277 (640), LYS293 (660), VAL297 (664) | _ | SER138 (505) H-donor | 18 |
| TSHR$_{368-764}$ MT2 | SER138 (505), LEU278 (645), SER274 (641), ALA277 (644), | - | VAL219 (586), ALA277 (644), LEU278 (645), LEU100 (467), VAL297 (664), PRO301(668), VAL219 (586) | _ | SER138 (505) H-donor | 14 |

The amino acid residues and their positions are designated as the three letter abbreviations and the corresponding number; in case of TSHR$_{368-764}$ the amino acid outside the first bracket indicates the position in predicted structure and the amino acid residues within first bracket indicates the real position in full structure, TSHR$_{1-764}$ protein; WT = Wild-type; MT1 = Mutant 1 (p.Ser508Leu) and MT2 = Mutant 2 (p.Glu727Asp).

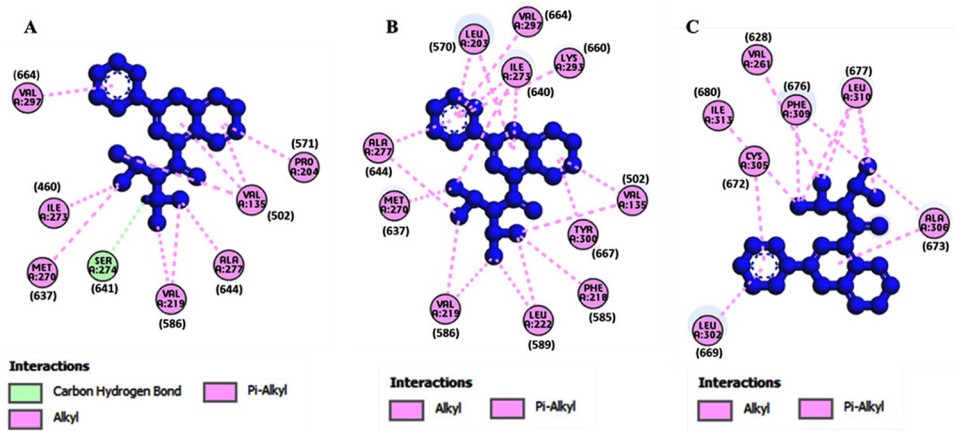

**Fig 4. Non-covalent interactions of MS437 with corresponding predicted structures of TSHR$_{368-764}$.** (A) TSHR$_{368-764}$ WT and MS437, (B) TSHR$_{368-764}$ MT1 and MS437 and (C) TSHR$_{368-764}$ MT2 and MS437.

MT2 (21739.906–24397.155Å$^2$). TSHR$_{368-764}$WT was unstable, but TSHR$_{368-764}$ MT1 and TSHR$_{368-764}$ MT2 were more stable as the ligand bound complexes in physiological condition.

The RMSF value deviated most for TSHR$_{368-764}$WT in between 1(368)-30(397) and 326 (693)-397(764) residues, where, TSHR$_{368-764}$ MT1 exhibited more fluctuations in 1(368)-47 (415) residues than TSHR$_{368-764}$ MT2. Overall, TSHR$_{368-764}$ MT2 was more stable as a complex than others due to least deviation in the whole run (Fig 6E). The total number of hydrogen bonds indicated structural rigidity of protein. In case of TSHR$_{368-764}$WT high frequency of hydrogen bonds (average ∼633) was observed during interaction while TSHR$_{368-764}$ MT1 exhibited average 593, and TSHR$_{368-764}$ MT2 manifested average 600 hydrogen bonds. Among the mutant structures, TSHR$_{368-764}$ MT2 displayed more structural stability in the simulation (Fig 6F). During simulation, for MS438 (Fig 7), the RMSD values of α-carbon atoms remained ∼5.251 Å in TSHR$_{368-764}$-WT, ∼5.53 Å in TSHR$_{368-764}$ MT1, and ∼5.39 Å in TSHR$_{368-764}$ MT2. The fluctuations had been observed in TSHR$_{368-764}$ MT1 and TSHR$_{368-764}$ MT2 during 30–50 ns while least was found in TSHR$_{368-764}$-WT. However, the average RMSD values of all the complexes were almost close (Fig 7A). The Rg manifested quite high deviations among

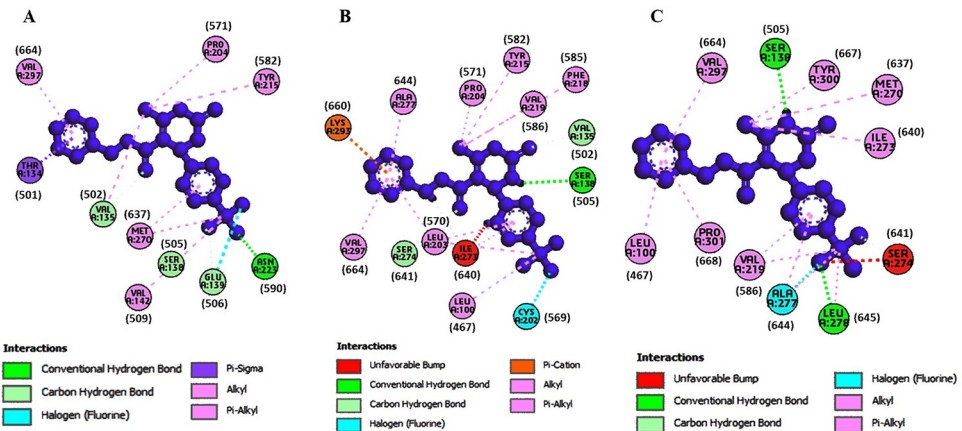

**Fig 5. Non-covalent interactions of MS438 with corresponding predicted structures of TSHR$_{368-764}$.** (A) TSHR$_{368-764}$ WT and MS438 (B) TSHR$_{368-764}$ MT1 and MS438 and (C) TSHR$_{368-764}$ MT2 and MS438.

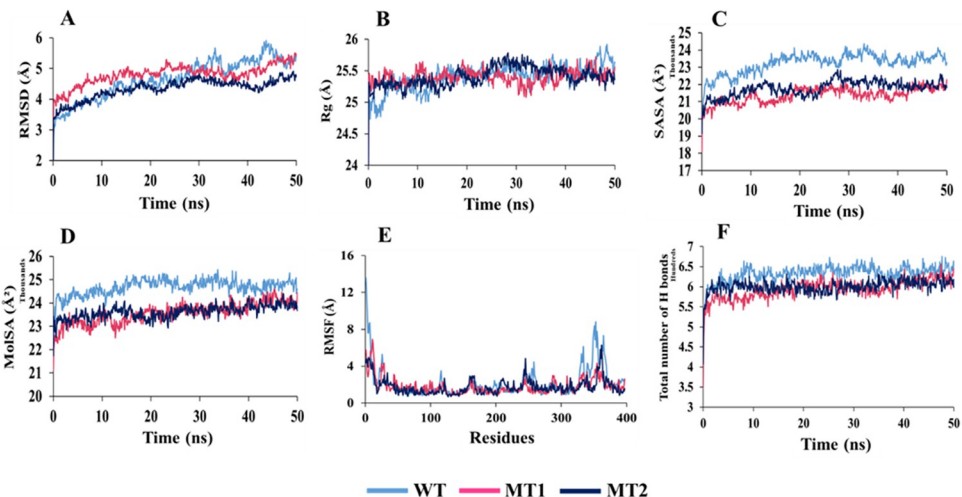

**Fig 6. Analysis of 50 ns MD simulation of TSHR$_{368-764}$ in complex with MS437 ligand.** (A) Root mean square deviation values of C-α atom. The structural changes of TSHR$_{368-764}$ proteins by means of (B) radius of gyration, (C) solvent accessible surface area, (D) molecular surface area, (E) root means square fluctuations, and (F) total number of hydrogen bonds formed during the simulation.

three complexes. However, TSHR$_{368-764}$ MT1 exhibited maximum 27.068 Å, which indicated higher stability than other complexes (Fig 7B). The SASA values remained close among three complexes. However, TSHR$_{368-764}$ WT manifested least deviations during 10–20 ns and 30–50 ns (Fig 7C). In case of MolSA, the graphical patterns for three complexes were almost same through the whole MD run (Fig 7D). The RMSF value more diverged in TSHR$_{368-764}$ MT2 till first 15 residues while least deviation was observed for TSHR$_{368-764}$ WT through whole run. However, three complexes showed quite similar pattern between 130(498)-240(608) residues (Fig 7E). The highest number of hydrogen bonds (about 678) was observed for TSHR$_{368-764}$ MT1 while TSHR$_{368-764}$ WT exhibited about 669 hydrogen bonds and TSHR$_{368-764}$ MT2 showed almost 665 to maintain stable conformation. Thus, TSHR$_{368-764}$MT1 showed highest

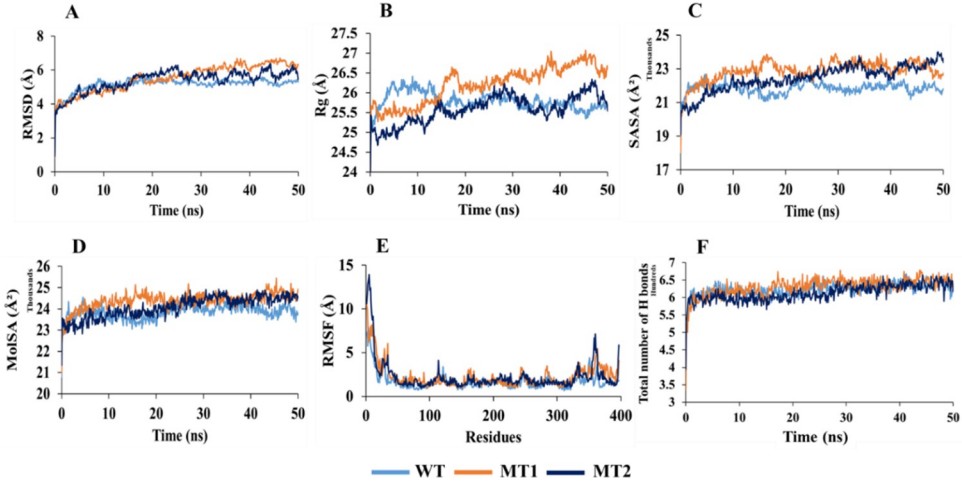

**Fig 7. Analysis of 50 ns MD simulation of TSHR$_{368-764}$ in complex with MS438 ligand.** (A) Root mean square deviation values of C-α atom. The structural changes of TSHR$_{368-764}$ proteins by means of (B) radius of gyration, (C) solvent accessible surface area, (D) molecular surface area, (E) root means square fluctuations, and (F) total number of hydrogen bonds formed during the simulation.

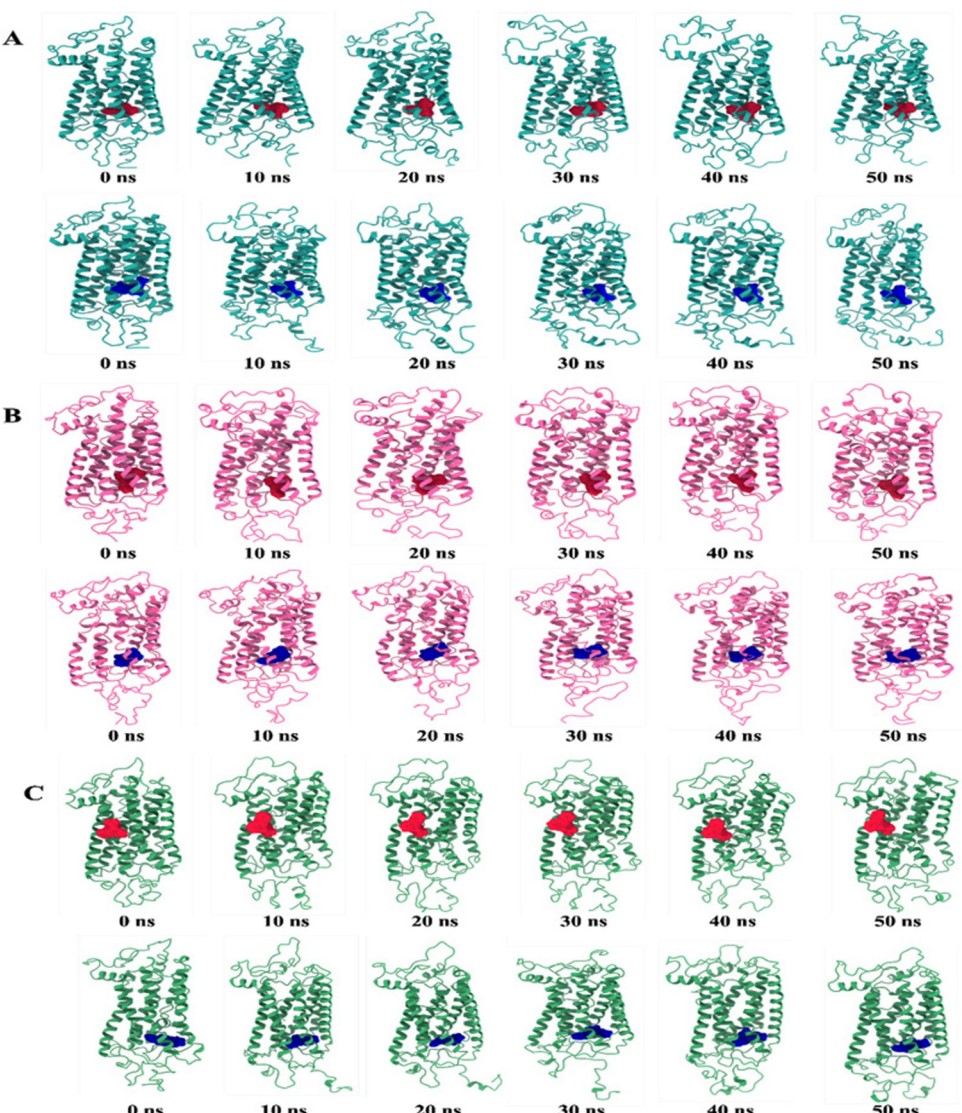

**Fig 8. The snapshots of the generated conformers for TSHR$_{368-764}$ and ligands: MS437 (red), MS438 (blue) over the 50 ns MD simulation.** (A) TSHR$_{368-764}$-WT (cyan), (B) TSHR$_{368-764}$-MT1 (pink), and (C) TSHR$_{368-764}$-MT2 (green) structures.

structural stability among the complexes (Fig 7F). Moreover, we had visualized the binding pattern of MS437 and MS438 ligands with the wild-type and mutants TSHR$_{368-764}$ through the snapshots from MD simulation (Fig 8). In simulation, MS437 exhibited persistent interaction with the residues LEU302(669), ALA306(673), LEU310(677) of TSHR$_{368-764}$MT2. The MS438 ligand mostly showed stable interactions with the residues LEU100(467), VAL135(502), SER138(505), LEU203(570), PRO204(571), LYS293(660). Both ligands remained within the binding site in stable mutant proteins.

### 3.5. Principal Component Analysis (PCA)

Two PCA models were generated for structural and energy profiles of the protein-ligand complexes to assess and realize the dissimilarities among wild-type and mutant proteins during

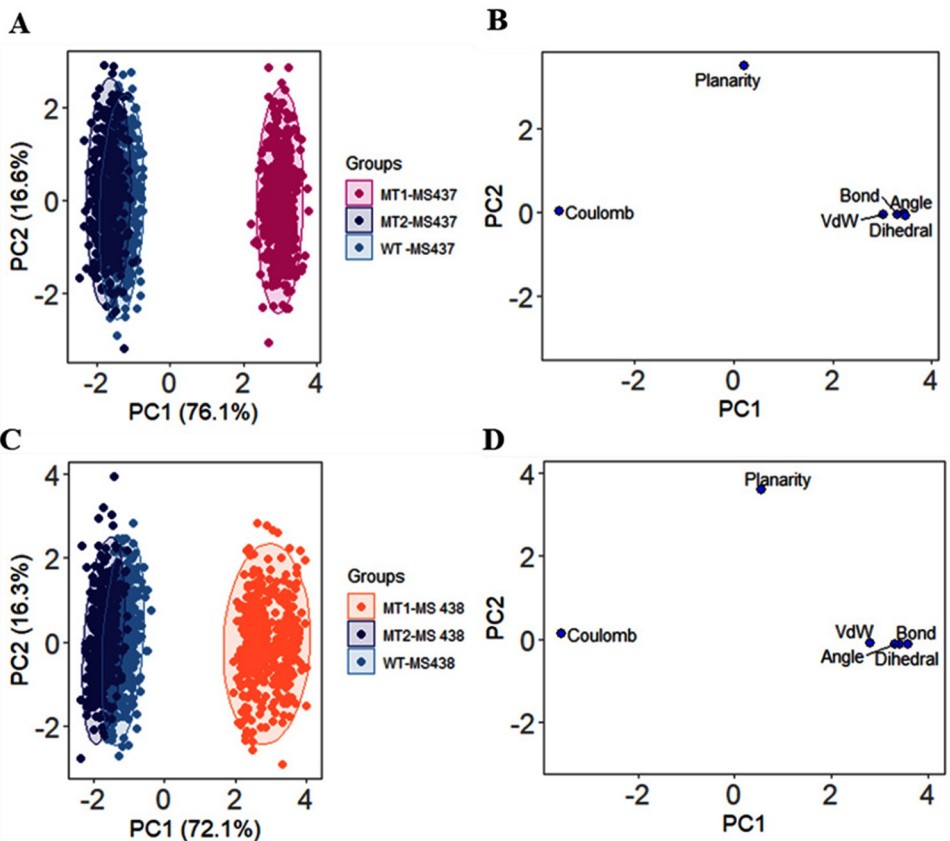

**Fig 9. PCA analysis on 50 ns MD simulation.** (A, C) The score plots represents three clusters for TSHR$_{368-764}$ wild-type and mutant protein structures, where each dot specifies one time point. The clustering is attributable as: WT-MS437 (sky blue), MT1-MS437 (pink), MT2-MS437 (navy blue), and WT-MS438 (sky blue), MT1-MS438 (orange), MT2-MS438 (navy blue), (B, D) Loading plots display the energy and structural profile data from principal component analysis.

MD simulation. The scores plot for MS437-protein (Fig 9A) and MS438-protein (Fig 9C) complexes had exhibited the different clusters for the wild-type and mutants of TSHR$_{368-764}$. It was observed that in both protein-ligand complexes, TSHR$_{368-764}$WT and TSHR$_{368-764}$MT1 were remotely situated. Consequently, pathogenic TSHR$_{368-764}$ MT1 was liable for the differences. However, TSHR$_{368-764}$WT and TSHR$_{368-764}$ MT2 were overlapped. The loading plots (Fig 9B and 9D) demonstrated that bond, bond angles, van der Waals energies, and dihedral angles were closely distributed and displayed quite similar graphical pattern. The distribution mainly contributed for PC1 variance while coulomb energy difference contributed to PC2 variance. In MS437-protein complexes, the total 92.7% of the variance had been unveiled by PC1 and PC2, where PC1 expressed 76.1% and PC2 expressed 16.6% of the variance. Moreover, in MS438-protein complexes, the total 88.3% of the variance had been disclosed by PC1 and PC2, where PC1 expressed 72.1% and PC2 expressed 16.3% of the variance.

## 4. Discussion

The newborn screening for endocrine disorders is not frequently practiced in Bangladesh. In this study, we focused on the etiology of dysgenesis types of Congenital Hypothyroid patients

having small glands or ectopic gland or agenesis (absent of thyroid gland). Different studies suggested that *TSHR* was the major gene responsible for growth and development of thyroid gland [2, 18]. TSH binds to the receptor and creates the signaling pathway through G-protein coupled-receptor and Cyclic AMP-mediated adenylate cyclase. The full-length protein structure of TSHR is still under investigation through crystallography. The available structures do not include the all-cytoplasmic residues. Mutations in the *TSHR* gene results from loss or gain of function of the protein that causes different phenotypic variations and lead to hyperthyrotopinemia to severe Congenital Hypothyroidism [18, 19]. Analysis of *TSHR* gene showed that two mutations were found, namely c.1523C>T and c.2181G>C in the patients and we analyzed the effect of mutations by using different bioinformatics tools. Almost all the tools such as Polyphen 2, Mutation Taster and PROVEAN were very much popular to analyze the mutational effect. The mutation c.1523C>T was found to be damaging, disease causing or deleterious and c.2181G>C was found to be benign or neutral. Since MS437 and MS438 are thyrogenic potent molecules, we selected these two molecules as ligands for molecular docking with the wild-type and mutant structures of TSHR protein [6]. The molecular docking analysis showed that the binding affinity for both of the ligands with mutant cases was decreased compared to the wild-type TSHR protein. MD simulation indicated that, the RMSDs for MS437-TSHR$_{368-764}$ MT2 (average 4.37Å) showed less deviations for α-carbon atoms. Thus, proposing the complex as most stable in biological environments. However, MS438-protein complexes manifested quite close average RMSD values during simulation. The Rg of MS437-TSHR$_{368-764}$WT exhibited more instabilities at start and end of the simulation. Conversely, TSHR$_{368-764}$ MT1 and TSHR$_{368-764}$ MT2 displayed quite similar pattern of lesser compactness as well as more stability for interaction with MS437. In case of MS438, TSHR$_{368-764}$ MT1 exhibited highest Rg value, which identified higher stability than other complexes. The analysis of SASA and MolSA values revealed that TSHR$_{368-764}$ MT1 and TSHR$_{368-764}$ MT2 mutant structures were more stable in their complex form with MS437 than TSHR$_{368-764}$WT. However, the SASA presented close pattern and MolSA exhibited almost same graphical pattern among the three complexes for the interaction with MS438. In case of hydrogen bonds, MS437-TSHR$_{368-764}$MT2 manifested average 600 hydrogen bonds which was close to TSHR$_{368-764}$WT (average $\sim$633). The complex was more stable than the other mutant. On the other hand, MS438-TSHR$_{368-764}$ MT1 displayed maximum structural stability compared to other complexes. Considering RMSF values, MS437 rendered more stability to TSHR$_{368-764}$MT2 than others. The RMSF value more diverged in TSHR$_{368-764}$MT2 while minimum deviance was detected for TSHR$_{368-764}$-WT and TSHR$_{368-764}$ MT1 mostly remained between both complexes. However, three complexes displayed almost similar stability between 130 (498)-240(608) residues while interacting with MS438. Moreover, PCA analysis for MS437-protein and MS438-protein complexes had revealed the existing differences among structural and energy profiles of the structures. It was observable that TSHR$_{368-764}$ MT1 exhibited much variations than TSHR$_{368-764}$WT and TSHR$_{368-764}$ MT2, emphasizing more damaging pattern in TSHR$_{368-764}$ MT1. In the study, we had utilized allosteric ligands MS437 and MS438 as agonists against the identified mutants for TSHR$_{368-764}$. These two ligands had 'drug-likeness' as well as previously confirmed their efficacy by conducting *in vivo* animal studies [6]. The agonists (MS437 and MS438) displayed different binding sites in the TSHR protein [6]. After analyzing all data, it can be proposed that low-affinity binding infers, a comparatively high concentration of the ligands can maximally occupy the binding sites to achieve maximum physiological response. Moreover, modifying chemical properties or ligands with novel scaffolds targeting signal-sensitive amino acids surrounding the allosteric binding sites might lead to design agonists with even higher efficiency to activate TSHR [19].

## 5. Conclusion

The study investigated the molecular etiology of thyroid dysgenesis. Sequencing-based analysis detected two mutation (p.Ser508Leu, p.Glu727Asp) in *TSHR* gene in Bangladeshi patients. The effect of mutations on TSHR protein was investigated targeting by small molecules drugs (MS437 and MS 438) via *in silico* approach using bioinformatics tools. The damaging effect in drug-protein complexes of mutants was revealed by molecular docking, non-covalent interaction, molecular dynamics simulation, and principle component analysis. The findings will be helpful to realize the molecular etiology of thyroid dysgenesis (TH) via exploring the mutational impact for TSHR protein and suggest more efficient treatment strategies including suitable drug design in future.

## Acknowledgments

The authors are grateful to the Institute for Developing Science and Health Initiatives (ideSHi), Bangladesh and Dr. Mohammad A. Halim, Division of Computer Aided Drug Design, The Red-Green Research Centre, Bangladesh.

## Author Contributions

**Conceptualization:** Mst. Noorjahan Begum, Kaiissar Mannoor, Sharif Akhteruzzaman, Firdausi Qadri.

**Data curation:** Mst. Noorjahan Begum, Rumana Mahtarin.

**Formal analysis:** Mst. Noorjahan Begum, Rumana Mahtarin, Md. Tarikul Islam, Sinthyia Ahmed.

**Investigation:** Mst. Noorjahan Begum, Rumana Mahtarin, Md. Tarikul Islam, Sinthyia Ahmed, Tasnia Kawsar Konika.

**Methodology:** Mst. Noorjahan Begum, Rumana Mahtarin, Sinthyia Ahmed.

**Supervision:** Kaiissar Mannoor, Sharif Akhteruzzaman, Firdausi Qadri.

**Visualization:** Mst. Noorjahan Begum.

**Writing – original draft:** Mst. Noorjahan Begum, Rumana Mahtarin.

**Writing – review & editing:** Mst. Noorjahan Begum, Rumana Mahtarin, Kaiissar Mannoor, Sharif Akhteruzzaman.

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
