## [Decision Letter · Decision Letter 0]

3 Apr 2023

PONE-D-23-04707Molecular Investigation of TSHR gene in Bangladeshi Congenital Hypothyroid patientsPLOS ONE

Dear Dr. Qadri,

Thank you for submitting your manuscript to PLOS ONE. After careful consideration, we feel that it has merit but does not fully meet PLOS ONE’s publication criteria as it currently stands. Therefore, we invite you to submit a revised version of the manuscript that addresses the points raised during the review process.

The reviewers raised a number of important concerns, including incomplete or misleading interpretation of the data. Once I receive a revised version of your manuscript, I will send it back to at least one of the original reviewers to determine its suitability for publication in PLOS ONE.

We look forward to receiving your revised manuscript.

Kind regards,

Dhanusha Yesudhas

Academic Editor

PLOS ONE

Journal Requirements:

Additional Editor Comments:

The manuscript require substantial modifications before considering for publication.

Please specify the region line 161, small molecule binding TM region.

The modeling of the TM region, author should consider using more than one computational tool to evaluate it, not only I-TASSER.

Since the whole work is relay on the model, the authors should have prepared the model from different tools subsequently compare and choose the most consistent one.

Table 4, write which one is acceptor and donor in proper format, the figure ligand is confusing, update it with mentioning of what are the numbers in braces. Same for table 5. It will be good to add one clear pic about the mutant position in the full length protein with more structural explanation (ref: https://www.ncbi.nlm.nih.gov/books/NBK279140/)

English editing is required.

Figure qualities need to be improved.

Reviewers' comments:

Reviewer's Responses to Questions

**Comments to the Author**

1. Is the manuscript technically sound, and do the data support the conclusions?

Reviewer #1: Yes

Reviewer #2: No

Reviewer #3: Partly

2. Has the statistical analysis been performed appropriately and rigorously? 

Reviewer #1: N/A

Reviewer #2: N/A

Reviewer #3: No

3. Have the authors made all data underlying the findings in their manuscript fully available?

Reviewer #1: Yes

Reviewer #2: Yes

Reviewer #3: Yes

4. Is the manuscript presented in an intelligible fashion and written in standard English?

Reviewer #1: No

Reviewer #2: No

Reviewer #3: Yes

5. Review Comments to the Author

Reviewer #1: Line 68-69 needs a little bit of reconstruction. Line 78, exons should be in small letters. In line no. 81,the sentence starts with an And. Line 82, what does normally placed thyroid gland mean? Line 85, how can two molecules at the same time have the highest potency? Line 139 is a bit confusing. line 150 Magnesium chloride 's formula is wrong.

Reviewer #2: Overall, the study has several limitations and shortcomings that raise questions about its originality, accuracy, and clinical relevance.

Lack of originality: The docking analysis has already been done by another study "DOI: 10.1089/thy.2014.0119", which raises questions about the originality and novelty of this study. Additionally, the identification of the mutations and their location in the protein have already been reported in the ClinVar database "NM_000369.5(TSHR):c.1523C>T (p.Ser508Leu), further reducing the novelty of this study.

Incorrect information: The authors have incorrectly identified the location of one of the mutations (Asp727Glu) in the transmembrane region of the protein, when in fact it is located in the cytoplasmic region. This suggests a lack of attention to detail and accuracy in the study.

Small sample size: The study only includes 21 patients with dysgenesis, which is a relatively small sample size. This may limit the generalizability and statistical power of the findings.

Reviewer #3: The manuscript “Molecular Investigation of TSHR gene in Bangladeshi Congenital Hypothyroid patients” reports two mutations in the transmembrane region of TSHR gene in 21 patients. The transmembrane region of TSHR proteins (wild type and mutant) are modeled using I-TASSER. The authors did docking and molecular dynamic simulations for the two small molecule drugs with the wild type as well as mutants to understand the effect of mutations. The authors used autodock vina for docking and YASARA suits for the MD simulations.

Overall application of different computational and modeling tools looks logical for the subject of the manuscript, the obtained results lack proper scientific analysis. The work is clearly presented, however I have several significant questions. The crude values are mentioned in the text (Table 3) rather than the necessary information about the target (TPO protein) and template (TSHR protein ) alignment. As a reader, I am unable to understand the quality of the predicted model based on the information provided. The full length protein is essential to understand the function. Why is the full length of TSHR protein not modeled? The alignment information should be added and discussed. How many models were predicted by the I-TASSER tool. At least the user should discuss the top three models and explain the differences or similarities. Where all the models used to be built by the same template. What is the quality of the template structure? It is easy to get automated protein structure using multiple tools. If the target template alignment is low , it will be interesting to compare the models built by another popular tool to see the difference in the prediction.When author discusses the C-score, it will be helpful to the reader to explain what the score means.Also, whether the small molecules were built or download from the database. The reference or web link for PyRx software can be added. There is no information about the active site nor about how many docking clusters for each of the small molecules with the protein. Are there any other docking poses found if the observed author can discuss those. Adequacy of modeling and docking were not also evaluated that is specially important for modeled proteins.

6. PLOS authors have the option to publish the peer review history of their article (what does this mean?). If published, this will include your full peer review and any attached files.

Reviewer #1: No

Reviewer #2: No

Reviewer #3: No

---

## [Author Response · Author response to Decision Letter 0]

17 May 2023

Responses to Reviewers’ Comments

Comment:1. Please ensure that your manuscript meets PLOS ONE's style requirements, including those for file naming. The PLOS ONE style templates can be found at 

Response: Thank you for your comment. We have revised the manuscript as PLOS ONE's style.

Comment 2. Thank you for stating the following financial disclosure: 

Comment (a) Please clarify the sources of funding (financial or material support) for your study. List the grants or organizations that supported your study, including funding received from your institution. 

Response: This study was partially funded by a grant from the University of Dhaka received from the University Grant Commission (CP-4029). 

Comment (b) State what role the funders took in the study. If the funders had no role in your study, please state: “The funders had no role in study design, data collection and analysis, decision to publish, or preparation of the manuscript.”

Response: The funders had no role in study design, data collection, and analysis, the decision to publish, or preparation of the manuscript

Comment (c) If any authors received a salary from any of your funders, please state which authors and which funders.

Response: The first author was a Ph.D. student (from 2016 to 2019) under a grant from the University of Dhaka. The Ph.D. program was finished in December 2019.

Comment (d) If you did not receive any funding for this study, please state: “The authors received no specific funding for this work.”

Response: The first author received fund for genomic sequence only during PhD program during 2016-2019. No fund was available for computational study. 

Response: Thank you. We have provided all the generated and analyzed data with the revised manuscript.

Response: Thank you. Orchid iD has been inserted.

 Additional Editor Comments:

The manuscript require substantial modifications before considering for publication.

Comment: Please specify the region line 161, small molecule binding TM region.

The modeling of the TM region, author should consider using more than one computational tool to evaluate it, not only I-TASSER.

Since the whole work is relay on the model, the authors should have prepared the model from different tools subsequently compare and choose the most consistent one.

Table 4, write which one is acceptor and donor in proper format, the figure ligand is confusing, update it with mentioning of what are the numbers in braces. Same for table 5. It will be good to add one clear pic about the mutant position in the full length protein with more structural explanation (ref: https://www.ncbi.nlm.nih.gov/books/NBK279140/)

English editing is required.

Figure qualities need to be improved.

Response: Thank you for your comment. The suggested corrections have been made accordingly and marked in the result section.

Reviewers' comments:

Reviewer's Responses to Questions

Comments to the Author

1. Is the manuscript technically sound, and do the data support the conclusions?

Reviewer #1: Yes

Reviewer #2: No

Reviewer #3: Partly

Our Response: Yes

2. Has the statistical analysis been performed appropriately and rigorously?

 Reviewer #1: N/A

Reviewer #2: N/A

Reviewer #3: No

Our Response: Yes

3. Have the authors made all data underlying the findings in their manuscript fully available?

Reviewer #1: Yes

Reviewer #2: Yes

Reviewer #3: Yes

Our Response: Yes

4. Is the manuscript presented in an intelligible fashion and written in standard English?

 Reviewer #1: No

Reviewer #2: No

Reviewer #3: Yes

Our Response: Yes

5. Review Comments to the Author

Reviewer #1: Comment: Line 68-69 needs a little bit of reconstruction. Line 78, exons should be in small letters. In line no. 81, the sentence starts with an And. Line 82, what does normally placed thyroid gland mean? Line 85, how can two molecules at the same time have the highest potency? Line 139 is a bit confusing. line 150 Magnesium chloride 's formula is wrong.

Response: Thank you for your comment. The suggested corrections have been made accordingly in the revised manuscript.

Reviewer #2: Comment: Overall, the study has several limitations and shortcomings that raise questions about its originality, accuracy, and clinical relevance.

Lack of originality: The docking analysis has already been done by another study "DOI: 10.1089/thy.2014.0119", which raises questions about the originality and novelty of this study. Additionally, the identification of the mutations and their location in the protein have already been reported in the ClinVar database "NM_000369.5(TSHR):c.1523C>T (p.Ser508Leu), further reducing the novelty of this study.

Response: Thank you for your comment. In Bangladesh, there was no genetic data are available on congenital hypothyroid patients with thyroid gland dysfunction. The present study aimed to investigate the mutation in TSHR gene and wanted to see the effect of mutation in possible drug binding site at the TSHR protein. We selected the drug and protein structure based on the above-mentioned article. However, investigating the mutational effect was unique in our study. We just investigated the mutation in Bangladeshi patients and wanted to see the effect of mutations on the drug-binding region in the TSHR protein. To do so, we have analyzed the binding affinity and interactions by molecular docking and structural and energy profiles of the protein-ligand complexes by molecular dynamics simulations followed by principle component analysis. Our intention was to explore the structure-function relationship of drugs within proteins by applying computational biology approaches. This study is important as mutation spectrum analysis helps to find out which drug is more suitable for treatment as well as to design a new drug for Congenital hypothyroidism. 

incorrect information: The authors have incorrectly identified the location of one of the mutations (Asp727Glu) in the transmembrane region of the protein, when in fact it is located in the cytoplasmic region. This suggests a lack of attention to detail and accuracy in the study.

Response: Thank you for your comment. The suggested corrections have been made accordingly.

Comment: Small sample size: The study only includes 21 patients with dysgenesis, which is a relatively small sample size. This may limit the generalizability and statistical power of the findings.

Response: We agree with you. the small sample size is our study limitation. However, the study was cross-sectional, we recruited patients from BSMMU for over 3 years and collected specimens from 2016 to 2019. The frequency of Congenital Hypothyroidism (CH) is 1 in 3000-4000 children worldwide and we enrolled a total of 65 participants (aged less than 18 years) which represents a total of 227,500 children in Bangladesh. Among them, 21 were found as thyroid dysgenesis and also subjected to investigation of mutation in the TSHR gene. The visits were performed twice a week. In the OPD (Outpatient Department) of Pediatrics Endocrinology and NINMAS, about 50-60 children visited per day with different endocrine disorders. Among them, 1 or 2 children were confirmed cases of Congenital Hypothyroidism who visited BSMMU for their follow-up examinations. Almost all the patients were late-diagnosed and kept under treatment for the Levothyroxine (LT4) drug. If we wanted to collect more samples for CH, then we had to visit more times and enrolled more populations, which was actually not possible in this study period.

Reviewer #3: Comment: The manuscript “Molecular Investigation of TSHR gene in Bangladeshi Congenital Hypothyroid patients” reports two mutations in the transmembrane region of TSHR gene in 21 patients. The transmembrane region of TSHR proteins (wild type and mutant) are modeled using I-TASSER. The authors did docking and molecular dynamic simulations for the two small molecule drugs with the wild type as well as mutants to understand the effect of mutations. The authors used autodock vina for docking and YASARA suits for the MD simulations.

Overall application of different computational and modeling tools looks logical for the subject of the manuscript, the obtained results lack proper scientific analysis. 

The work is clearly presented, however I have several significant questions. The crude values are mentioned in the text (Table 3) rather than the necessary information about the target (TPO protein) and template (TSHR protein) alignment. As a reader, I am unable to understand the quality of the predicted model based on the information provided. 

Response: Thank you for your comment. The alignment information has been added in the method section. The details about the accuracy and quality of the predicted model have been provided also.

Comment: The full-length protein is essential to understand the function. Why is the full length of TSHR protein not modeled? 

Response: Thank you for your comment. Initially, the full-length protein was not modeled due to a lack of full-length experimental structure. However, now the structure is modeled using AlphaFold and superimposed the transmembrane region and found similar results. 

Comment: The alignment information should be added and discussed. How many models were predicted by the I-TASSER tool. At least the user should discuss the top three models and explain the differences or similarities. 

Response: Thank you for your comment. Yes, we have explained the top model from the given five models for each protein, we excluded the rest due to low quality. We have explained in detail the accuracy and quality of the predicted models and marked them for your consideration. 

Comment: Where all the models used to be built by the same template. What is the quality of the template structure? It is easy to get automated protein structure using multiple tools. If the target template alignment is low, it will be interesting to compare the models built by another popular tool to see the difference in the prediction.

Response: Thank you for your comment. We have compared the I-TASSER predicted structure with AlphaFold predicted TSHR model protein.

Comment: When author discusses the C-score, it will be helpful to the reader to explain what the score means.Also, whether the small molecules were built or download from the database. The reference or web link for PyRx software can be added. There is no information about the active site nor about how many docking clusters for each of the small molecules with the protein. Are there any other docking poses found if the observed author can discuss those. Adequacy of modeling and docking were not also evaluated that is specially important for modeled proteins.

Response: Thank you for your comment. The suggested corrections have been made accordingly and marked in the result section.

6. PLOS authors have the option to publish the peer review history of their article (what does this mean?). If published, this will include your full peer review and any attached files.

Do you want your identity to be public for this peer review? For information about this choice, including consent withdrawal, please see our Privacy Policy.

Response: Yes

---

## [Decision Letter · Decision Letter 1]

21 Jun 2023

PONE-D-23-04707R1Molecular Investigation of TSHR Gene in Bangladeshi Congenital Hypothyroid patientsPLOS ONE

Dear Dr. Qadri,

Thank you for submitting your manuscript to PLOS ONE. After careful consideration, we feel that it has merit but does not fully meet PLOS ONE’s publication criteria as it currently stands. Therefore, we invite you to submit a revised version of the manuscript that addresses the points raised during the review process.

We look forward to receiving your revised manuscript.

Kind regards,

Dhanusha Yesudhas

Academic Editor

PLOS ONE

Additional Editor Comments:

The authors should pay more attention about the revised copy of the manuscript, the tracked word file is difficult to read and manage. Please make a clean copy word file highlighting the modifications.

Various part of the manuscript remains hard to read and lack of connection.

The authors mentioned about the addition of figures, but the figure remains same. Please update it accordingly.

Reviewers' comments:

Reviewer's Responses to Questions

**Comments to the Author**

1. If the authors have adequately addressed your comments raised in a previous round of review and you feel that this manuscript is now acceptable for publication, you may indicate that here to bypass the “Comments to the Author” section, enter your conflict of interest statement in the “Confidential to Editor” section, and submit your "Accept" recommendation.

Reviewer #3: (No Response)

2. Is the manuscript technically sound, and do the data support the conclusions?

Reviewer #3: Partly

3. Has the statistical analysis been performed appropriately and rigorously? 

Reviewer #3: I Don't Know

4. Have the authors made all data underlying the findings in their manuscript fully available?

Reviewer #3: No

5. Is the manuscript presented in an intelligible fashion and written in standard English?

Reviewer #3: No

6. Review Comments to the Author

Reviewer #3: The writing part should be improved. Please improve the clarity and explain the necessary information in detail so that the paper explains independently. As a reader I am not convinced the author's writings are clear yet.

-What was the target template alignment percentage? Is it 30% or 70%? If the target template is very low, authors can submit distance guided protein structure https://zhanggroup.org/D-I-TASSER/. Nowadays, getting automated protein models in a few hours. Again, the authors didn’t mention the target template hence I cannot decide.

-Additional figures would have explained the target template alignment. e.g. Like the I-TASSER help page suggests https://zhanggroup.org/I-TASSER/example/ . The authors can add the figure to show the target template alignment (Page 47 for the section Prediction of 3D structure of TSHR protein and ligand selection).

- As pointed out by the other reviewer the author’s lack attention to detail. For instance in the revised manuscript the reference 8 points out to the I-Tasser reference instead of the previous high throughput study.(Introduction, Page 44, line 111).

- The author’s may submit the predicted protein and docking models in a public repository like https://figshare.com/ that might be useful to other researchers

7. PLOS authors have the option to publish the peer review history of their article (what does this mean?). If published, this will include your full peer review and any attached files.

Reviewer #3: No

---

## [Author Response · Author response to Decision Letter 1]

10 Jul 2023

Additional Editor Comments:

The authors should pay more attention about the revised copy of the manuscript, the tracked word file is difficult to read and manage. Please make a clean copy word file highlighting the modifications.

Various part of the manuscript remains hard to read and lack of connection.

The authors mentioned about the addition of figures, but the figure remains same. Please update it accordingly.

Response: Thank you for your comment. The suggested corrections have been made accordingly

Reviewers' comments:

Reviewer's Responses to Questions

Comments to the Author

1. If the authors have adequately addressed your comments raised in a previous round of review and you feel that this manuscript is now acceptable for publication, you may indicate that here to bypass the “Comments to the Author” section, enter your conflict of interest statement in the “Confidential to Editor” section, and submit your "Accept" recommendation.

Reviewer #3: (No Response)

Our Response: Yes

2. Is the manuscript technically sound, and do the data support the conclusions?

Reviewer #3: Partly

Our Response: Yes

3. Has the statistical analysis been performed appropriately and rigorously?

Reviewer #3: I Don't Know

Our Response: Yes

4. Have the authors made all data underlying the findings in their manuscript fully available?

Reviewer #3: No

Our Response: Yes. Additionally, we have shared the predicted proteins and docking models in the public repository, https://figshare.com/. 

Here is the link: https://figshare.com/s/b10e48d2a8e110cca57e

5. Is the manuscript presented in an intelligible fashion and written in standard English?

Reviewer #3: No

Our Response: Yes

6. Review Comments to the Author

Reviewer #3: Comment: The writing part should be improved. Please improve the clarity and explain the necessary information in detail so that the paper explains independently. As a reader I am not convinced the author's writings are clear yet.

What was the target template alignment percentage? Is it 30% or 70%? If the target template is very low, authors can submit distance guided protein structure https://zhanggroup.org/D-I-TASSER/. Nowadays, getting automated protein models in a few hours. Again, the authors didn’t mention the target template hence I cannot decide.

Additional figures would have explained the target template alignment. e.g. Like the I-TASSER help page suggests https://zhanggroup.org/I-TASSER/example/ . The authors can add the figure to show the target template alignment (Page 47 for the section Prediction of 3D structure of TSHR protein and ligand selection).

Response: Thank you for your comment. We have substantially improved the writing. Target template alignments were above 60% (62%, 64%, and 63%) so we did not obtain structures from distance guided protein structure https://zhanggroup.org/D-I-TASSER/ . We have clearly explained the target template alignments in section 2.4. and 3.2. and additional figures have been shown for the target template alignments.

Comment: As pointed out by the other reviewer the author’s lack attention to detail. For instance in the revised manuscript the reference 8 points out to the I-Tasser reference instead of the previous high throughput study. (Introduction, Page 44, line 111).

Response: Thank you for your comment. The suggested corrections have been made accordingly in the revised manuscript.

Comment: The author’s may submit the predicted protein and docking models in a public repository like https://figshare.com/ that might be useful to other researchers

Response: Thank you for your comment. We have submitted the predicted proteins and docking models in a public repository https://figshare.com/

Here is the link: https://figshare.com/s/b10e48d2a8e110cca57e

7. PLOS authors have the option to publish the peer review history of their article (what does this mean?). If published, this will include your full peer review and any attached files.

Response: Yes

---

## [Editor Report · Decision Letter 2]

12 Jul 2023

Molecular Investigation of TSHR Gene in Bangladeshi Congenital Hypothyroid patients

PONE-D-23-04707R2

Dear Dr. Qadri,

Thank you for submitting your work to PLOS ONE. I am happy to inform you that your paper "Molecular Investigation of TSHR Gene in Bangladeshi Congenital Hypothyroid patients" has been accepted for publication.

Regards,

Dhanusha Yesudhas

Academic Editor

PLOS ONE

Additional Editor Comments (optional):

Please look for the font changes, table alignment and figure resolutions before publication.
---

## [Editor Report · Acceptance letter]

3 Aug 2023

PONE-D-23-04707R2 

Molecular Investigation of *TSHR* Gene in Bangladeshi Congenital Hypothyroid Patients 

Dear Dr. Qadri:

I'm pleased to inform you that your manuscript has been deemed suitable for publication in PLOS ONE. Congratulations! Your manuscript is now with our production department. 

Kind regards, 

on behalf of

Dr. Dhanusha Yesudhas 

Academic Editor

PLOS ONE